# Glutathione Blood Concentrations: A Biomarker of Oxidative Damage Protection during Cardiopulmonary Bypass in Children

**DOI:** 10.3390/diagnostics9030118

**Published:** 2019-09-13

**Authors:** Angela Satriano, Simone Franchini, Giuseppe Lapergola, Francesca Pluchinotta, Luigi Anastasia, Ekaterina Baryshnikova, Giovanni Livolti, Diego Gazzolo

**Affiliations:** 1Department of Pediatric Cardiac Surgery, IRCCS San Donato Milanese Hospital, San Donato Milanese, 20097 Milan, Italy; 2Neonatal Intensive Care Unit, G. d’Annunzio University of Chieti, 65100 Chieti, Italy; 3Department of Biomedical and Biotechnological Sciences Section of Biochemistry University of Catania, 95100 Catania, Italy; 4AO SS Antonio, Biagio and C. Arrigo Hospital Alessandria, 15121 Alessandria, Italy

**Keywords:** GSH, cardiopulmonary bypass, newborn, brain damage, oxidative stress

## Abstract

Background. Pediatric open-heart surgery with cardiopulmonary bypass (CPB) still remains a risky interventional procedure at high mortality/morbidity. To date, there are no clinical, laboratory, and/or monitoring parameters providing useful information on perioperative stress. We therefore investigated whether blood concentrations of glutathione (GSH), a powerful endogenous antioxidant, changed in the perioperative period. Methods. We conducted an observational study in 35 congenital heart disease (CHD) children in whom perioperative standard laboratory and monitoring parameters and GSH blood levels were assessed at five monitoring time points. Results. GSH showed a pattern characterized by a progressive increase from pre-surgery up to 24 h after surgery, reaching its highest peak at the end of CPB. GSH measured at the end of CPB correlated with CPB duration, cross-clamping, arterial oxygen partial pressure, and with body core temperature. Conclusions. The increase in GSH levels in the perioperative period suggests a compensatory mechanism to oxidative damage during surgical procedure. Caution is needed in controlling different CPB phases, especially systemic reoxygenation in a population that is per se more prone to oxidative stress/damage. The findings may point the way to detecting the optimal temperature and oxygenation target by biomarker monitoring.

## 1. Introduction

Pediatric open-heart surgery with cardiopulmonary bypass (CPB) still remains a risky interventional procedure at high mortality and morbidity [1]. Hemodynamic and thermal changes occurring during CPB are known to trigger a cascade of events (i.e., ischemia-reperfusion injury, surgical procedure, endothelial dysfunction, activation of complement, coagulation, and inflammatory processes) that can lead to tissue damage [2].

The assessment of proinflammatory cytokines and intracellular biomarkers has been recently proposed in order to offer useful information on tissue stress in the perioperative period [3]. In this regard, it has been shown that glutathione (GSH) is essential for vascular and cardiac function and determines cell survival [4,5,6,7]. Moreover, in humans and in animal heart failure models, exacerbated tumor necrosis factor and soluble tumor necrosis factor receptor-1 expression was related to systemic and cardiac GSH deficiency supporting the notion of GSH’s role in the defense against oxidative stress [4,5,8,9,10].

GSH has been detected in different biological fluids and it has been suggested that its presence provides a good source of this protective antioxidant for newborns. GSH performs several important physiological functions, such as (a) inactivation of oxygen-derived highly reactive species [10,11]; (b) detoxification of various types of xenobiotics and carcinogens [10,12]; (c) maintenance of the oxidative status of other antioxidants, such as ascorbic acid and α-tocopherol [13]; and (d) cell immune response improvement by activation of lymphocytes [14]. In this light, GSH in adults undergoing surgical repair with CPB has been shown to increase, reflecting a cellular defense mechanism against oxidative damage during reperfusion [15]. Conversely, data on GSH concentrations in children, affected by congenital heart disease (CHD) and undergoing cardiac surgery, are still lacking or are a matter of debate. Reduced GSH levels has been recently shown to play an important role in inflammatory response [3], whereas changes in GSH blood levels were observed according to different CPB phases (i.e., controlled anterograde low oxygen warm reperfusion before de-clamping and aortic cross-clamping release) [15].

Therefore, in the present observational study we investigated whether, in CHD children undergoing cardiac surgery with CPB, blood concentrations of a powerful endogenous antioxidant, namely GSH, changed in the perioperative period.

## 2. Materials and Methods

We conducted an observational study at our third-level referral center for pediatric cardiac diseases of 35 CHD children without pre-existing neurological disorders or other co-morbidities. CHD characteristics as well as main interventions and clinical laboratory parameters recorded at admission into the study are reported in Table 1.

Informed and signed consent from parents was obtained, before the patients’ inclusion in the study, which was approved by the local human investigation committee (N718, EC Policlinico San Donato Milanese, 12, 12, 2012).

At admission to our unit, all children underwent clinical and standard laboratory and monitoring parameter recordings and GSH assessment. Blood samples were drawn at five predetermined times before, during, and after surgery, namely, before the surgical procedure and anesthesia (time 0, T0); during the surgical procedure after sternotomy and before CPB (time 1, T1); at the end of CPB (time 2, T2); at the end of the surgical procedure (time 3, T3); 24 h after the surgical procedure (time 4, T4). The following parameters were also recorded: peripheral temperature; nasopharyngeal temperature; pump flow rate; heart rate (HR) mean arterial systolic and diastolic blood pressure (BP); left and right atrium BP (LA, RA); pulsed arterial oxygen tension (SaO_2_); and laboratory parameters such as arterial blood pH and oxygen (PaO_2_) and carbon dioxide (PaCO_2_) partial pressures, bicarbonate (HCO_3_), base excess (BE), and hemoglobin (Hb) concentrations, hematocrit rate (Ht), platelet count, and creatinine, urea, lactate dehydrogenase, creatine kinase, and glucose blood levels.

### 2.1. Anesthetic Technique

After premedication with midazolam (Ipnovel, Roche, Milan, Italy), 0.5 mg/kg·bw (intramuscular), induction was achieved with oxygen and 3% sevofluorane (SevoFlo, Zoetis Belgium SA, Louvain-la-Neuve, Belgium) administered via mask (single-breath induction), followed by intravenous sufentanil (Fentanest, Pharmacia and Upjohn, Milan, Italy) 1 (g/kg·bw) and vecuronium (Norcuron, NV organon, Oss, The Netherlands) (0.15 mg/kg·bw). Maintenance was achieved with 3% sevofluorane (except during CPB) and with additional doses of sufentanil (0.5 g/kg·bw) and vecuronium (0.1 mg/kg·bw) every 30–40 min. During CPB, in the absence of sevofluorane, additional midazolam at 0.2 mg/kg·bw dosage was given. Sufentanil infusion at 0.25 g/kg·bw was continued in the intensive care unit for sedation [16].

### 2.2. Cardiopulmonary Bypass Management

The study population underwent a CPB procedure according to our protocols [16]. CPB was established after systemic heparinization (3 mg/kg·bw) by standard single-stage aortic and bicaval cannulation, and maintained via non-pulsatile pump flow with a membrane oxygenator (Dideco Laboratories, Modena, Italy). Flow velocity was kept at 120–150 mL/kg·bw, and mean arterial blood pressure at 45 mmHg; hypothermia was attained by core and surface cooling. Three patients were operated in deep hypothermic circulatory arrest (DHCA) (19 ± 7 min) and the minimum temperature reached was 25 °C. Mean rewarming time was calculated from the final temperature during hypothermic circulatory arrest to 36.5 °C. The pump priming solution was composed of electrolyte solutions (Normosol-R 250 to 650 mL, Abbott Hospital Products, Abbott Park, IL, USA or Plasma-Lyte A, Travenol Laboratories, Inc., Deerfield, IL, USA), albumin (25%), heparin 1000 to 5000 units in the total solution, sodium bicarbonate (25–30 mEq/L), and packed red blood cells or fresh frozen plasma. A standard total circuit prime volume was used, according to body weight, varying from 400 mL (bw < 4.5 kg), to 600 mL (bw > 4.5 kg and bw < 7.5 kg) and to 700 mL (bw > 7.7 kg). Packed red blood cells (200 to 500 mL) were transfused as needed to maintain a hematocrit level above 30% during CPB. Protamine (1 mg for each mg of heparin) was administered at the end of CPB. The α-stat regimen was used, and the PaCO_2_ maintained between 35 and 40 mmHg, without mathematical correction for the effects of the temperature, by varying the membrane oxygenator gas flow [16].

Modified ultrafiltration (MUF) was routinely performed before removal of arterial and venous cannulae. In the CPB circuit, the arterial line was connected to the inlet and the venous line to the outlet of the ultrafilter. As the patient was separated from the CPB, the clamp was removed from the inlet of the filter, allowing the blood to flow through the arterial line to the filter (10–15 mL/kg/min) and, finally, from a venous line to the RA. The filter allows the passage of molecules smaller than 65 kD molecular weight. When it was necessary to maintain the intravascular volume and stabilize the hemodynamics, the blood returned via the venous reservoir and the venous cannula to the RA. This technique was performed until the Ht achieved the target of 35% [17].

### 2.3. GSH Measurement

Levels of nonproteic thiol groups were measured in 200 μL of blood, in accordance with the method presented by Hu [18], with partial modifications. This spectrophotometric assay is based on the reaction of thiol groups with 2,2-dithio-bis-nitrobenzoic acid (DTNB) in absolute ethanol to give a colored compound absorbing at λ = 412 nm. Since the DTNB method is strongly affected by pH, the possibility of avoiding acids (trichloroacetic or sulfosalicylic acid) to precipitate proteins represents a remarkable advantage to ensure the accuracy of the assay. We then carried out the removal of proteins with an excess of absolute ethanol, followed by centrifugation at 3000*g* for 10 min at room temperature. Each value represents the mean ± SD of three experimental determinations for each sample. Results were expressed as micromoles per milliliter of blood.

### 2.4. Neurological Follow-up

Neurological examination was performed daily according to Prechtl [19]. Each child was assigned to one of three diagnostic groups—normal, suspect, abnormal. A child was considered to be abnormal when one or more of the following neurological syndromes were unequivocally present: (a) increased or decreased excitability (hyperexcitability syndrome, convulsion, apathy syndrome, coma); (b) increased or decreased motility (hyperkinesia, hypokinesia); (c) increased or decreased tonus (hypertonia, hypotonia); (d) asymmetries (peripheral or central); (e) defects of the central nervous system; (f) any combination of the above. When indications of the presence of a syndrome were non-conclusive or if only isolated symptoms were present (e.g., mild hypotonia or only a slight tremor), the children were classified as suspect. Abnormal and suspect cases were excluded from the study.

### 2.5. Statistical Analysis

For the calculation of sample size, we used GSH changes as the main parameter. As no basic data are available for a studied population, we assumed a decrease of 0.5 SD in GSH to be clinically significant. Indeed, considering α = 0.05 and using a two-sided test, we estimated a power of 0.80 recruiting 33 CHD patients. We added *n* = 2 cases to allow any dropout. The sample size was calculated by using nQuery Advisor (Statistical Solutions, Saugus, MA, USA), version 5.0. The Kolmogorov–Smirnov test showed values having a Gaussian distribution, and data were expressed as the mean (SD). Statistical significance was assessed using one-way ANOVA for repeated measures (followed by the post-hoc Tukey test for multiple comparisons) and the unpaired *t*-test when only two groups were compared. Linear regression analysis was used for correlation between GSH and CPB and laboratory parameters. Statistical significance was set at *p* < 0.05.

## 3. Results

Monitoring and laboratory variables at the predetermined perioperative time points are shown in Table 2. Clinical parameters recorded during surgery remained within limits regarded as within the reference ranges during this type of procedure. No significant differences from T0 to T4 (*p* > 0.05, for all) were observed regarding hemoglobin concentration; hematocrit rate; arterial blood pH, PaCO_2_, HCO_3_, BE, and SaO_2_; HR; LA BP; RA BP; systolic and diastolic BP; and glucose blood levels. PaO_2_ showed a pattern, from T0 to T4, characterized by a significant (*p* < 0.01, for all) progressive increase in concentration reaching its highest peak at T2 (*p* < 0.001) and progressively decreasing at the end of surgery up to 24 h after the surgical procedure (Table 2).

No overt neurological injury was observed in surviving patients during the first week after surgery. In CHD children, GSH blood levels were measurable at all monitoring time points. GSH showed a pattern characterized by a progressive increase from T0 (before surgery) to T4 (at 24 h after surgery), reaching its highest peak at T3 (at the end of CPB) and remaining at high levels up to T4 (*p* < 0.01, for all) (Figure 1).

No significant differences (*p* > 0.05, for both) in GSH levels were shown from T0 to T2. Moreover, in order to evaluate the potential side effects of PaO_2_ on GSH levels, we calculated the PaO_2_/GSH ratio. The PaO_2_/GSH pattern was characterized by a significant (*p* < 0.05, for all) increase from T0 to T3, whereas no differences (*p* > 0.05) were found between T0 and T4.

Linear regression analysis showed a significant correlation between GSH levels measured at the end of CPB (T2). In particular, a positive significant correlation was found with CPB duration (R = 0.72; *p* < 0.001), with cross-clamping (R = 0.62, *p* < 0.001), and with PaO_2_ (R = 0.53; *p* = 0.002), whereas a negative significant correlation was observed with body core temperature (R = −0.49; *p* = 0.012) (Figure 2, Figure 3 and Figure 4).

Moreover, an identical pattern was found when PaO_2_/GSH was correlated with CPB duration (R = 0.56; *p* = 0.004), with cross-clamping (R = 0.51, *p* < 0.010), and with body core temperature (R = 0.68; *p* < 0.001).

## 4. Discussion

It is widely known that congenital heart diseases are the most frequent malformations, affecting 1–20 infants in 1000 live births [1]. According to CHD characteristics, they may account for up to 15% of mortality, as well as about 4.8% of short-term and 40–50% of long-term developmental abnormalities at school age [1,20,21]. However, there are still no conclusive clinical, laboratory, or monitoring parameters able to provide useful information on perioperative stress [22,23,24,25]. In this setting, open-heart surgery and CPB are commonly accepted to cause harmful effects on whole body organs. Vascular resistance impairment, intra/postoperative hypoperfusion leading to ischemia reperfusion injury, and systemic inflammation are known to enhance an oxidative stress reaction in the perioperative period [26].

In the present study, we found that the blood levels of a well-established marker of oxidative stress, namely GSH, significantly increased during surgical procedure by CPB, remaining constantly elevated up to 24 h after surgery. Furthermore, GSH levels measured at the end of the CPB procedure positively correlated with CPB, cross-clamp duration, and PaO_2_, whereas a negative correlation was observed with body core temperature. These results are, in part, in agreement with previous observations in adults and in children [15,27], showing increased GSH levels in the postoperative period and in the different CPB phases. Discrepancies are in relation to the different monitoring time points, CPB procedure, and patient recruitment [15].

The finding of elevated GSH levels in the perioperative period warrants further consideration. In particular, increased GSH levels during CPB procedure are (1) related to enhanced oxidative damage due to hypoxia or hyperoxia insults during CPB itself [2]; (2) correlated with the complexity of surgical procedures, especially in those cases requiring DHCA [23]; (3) temperature-dependent due to changes in the oxygen–hemoglobin dissociation curve, enhancing hemoglobin affinity for oxygen with an increase in reduced/oxidized glutathione ratio [28]; and, (4) related to the use of hyperoxic anesthesia that stimulates the production of reactive nitrogen and oxygen species (RNOS) by neutrophils and mitochondria [29].

Altogether, it is reasonable to conclude that the increased release of GSH into systemic circulation in CHD children who had undergone open heart surgery can be considered as a compensatory mechanism to oxidative damage. The fact is of relevance bearing in mind that GSH is implicated the most in the protection against oxidative injury in glia, astrocytes, and neurons [30]. The mechanism through which GSH exerts its antioxidant action is, at this stage, not fully elucidated. One explanation resides in hypoxia or hyperoxia insults able to trigger the cellular GSH content, increasing the effect of N-acetylcysteine on repletion of GSH and further increasing key enzyme activities that regulate GSH production, glutamate cysteine ligase, and glutathione synthase [31,32]. In this light, it is reasonable to suggest that high GSH levels in blood may be the expression of a cascade of events affecting the glutathione/glutathione-oxidized ratio, which is hypoxia or hyperoxia mediated. Of note, high GSH levels in systemic circulation enhance its antioxidant capacity, thus reducing the proteins’ defense mechanisms by oxidative damage. Another explanation resides in therapeutic strategies performed in the perioperative period such as transfusions (red blood cells, platelets), which contribute to redox imbalance being an important source of RNOS [33,34]. Finally, the possibility that therapeutic strategies such as anesthesia and fluid balance management can somewhat affect GSH levels must be taken into account. The fact is of relevance bearing in mind that (i) perioperative therapeutic strategies performed did not contain any thiol group, and (ii) fluid management was not responsible for any hemodilution/concentration phenomenon. Since no significant differences in Hb and Ht rate were observed at monitoring time points, these findings argue against the hypothesis.

In the present study, we also found that GSH increases in the perioperative period as PaO_2_ peaks. Therefore, in order to limit the potential influence of oxygenation on GSH levels, we calculated the PaO_2_/GSH ratio. The PaO_2_/GSH ratio pattern increased during surgery up to the end of the procedure when it started to decrease, being superimposable to preoperative levels. The PaO_2_/GSH ratio showed a positive correlation with CPB duration, cross-clamping, and body core temperature.

The finding of a correlation between GSH and PaO_2_ deserves further consideration. In particular, increased GSH has shown the following: (i) in animal models, controlled normoxic reoxygenation limited brain injury [35]; (ii) in CHD children, lipid peroxidation increased and antioxidant reserve capacity decreased, being maintained by endogenous radical scavengers such as GSH [35,36,37,38,39,40]; and (iii) during CPB, increased oxygen toxicity affected endothelial permeability [2,35]. Altogether, it is reasonable to suggest that the higher GSH levels occurring during systemic reoxygenation in CHD children may be indicative of a compensatory mechanism due to CPB and hyperoxia-mediated stress.

Lastly, we found that the PaO_2_/GSH ratio was superimposable to the one detected at the preoperative time point. The explanation may reside in the decay of PaO_2_ levels at 24 h after surgery and explained by the short half-life of GSH (about 10 min) [41].

Finally, the present study shows some limitations. The main ones reside in the small population studied that did not allow to correct for the occurrence of cyanotic and non-cyanotic CHD and in the lack of an expanded oxidative stress evaluation with other biomarkers such as glutathione disulfide, non-protein bind iron and lipid hydroperoxides. Further studies in this regard are awaited.

In conclusion, the present data on elevated GSH levels, in CHD children undergoing open heart surgery and CPB, suggest that caution is needed in controlling different CPB phases such as DHCA, core body temperature, and systemic reoxygenation in a population that is per se more prone to CNS stress or damage. The findings may lead to further investigation to detect the “optimal” temperature and PaO_2_ target [40] by monitoring biomarkers as much as possible to avoid perioperative CNS stress or damage.

## Figures and Tables

**Figure 1 diagnostics-09-00118-f001:**
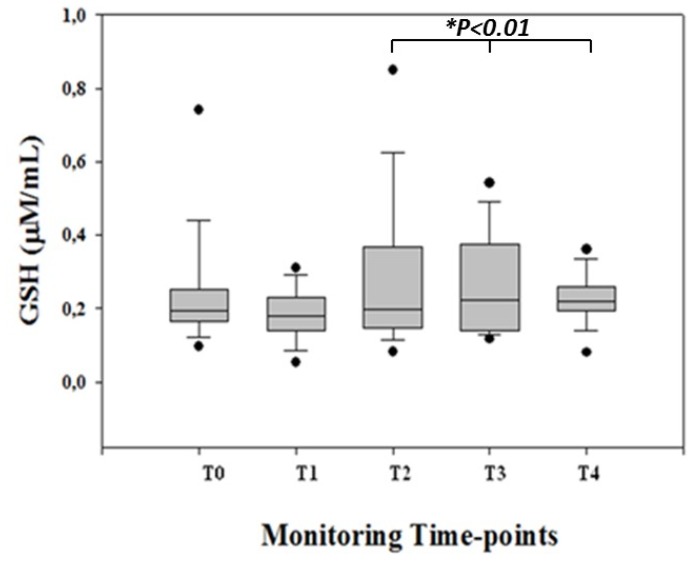
GSH concentrations (µM/mL) expressed as median and 5°–95° centiles at different monitoring time points: before the surgical procedure (T0); during the surgical procedure after sternotomy and before CPB (T1); at the end of CPB (T2); at the end of the surgical procedure (T3); and at 24 h after the surgical procedure (T4) in congenital heart disease children. GSH was significantly (**p* < 0.01, for all) higher at T3 and T4 vs. T0 monitoring time points.

**Figure 2 diagnostics-09-00118-f002:**
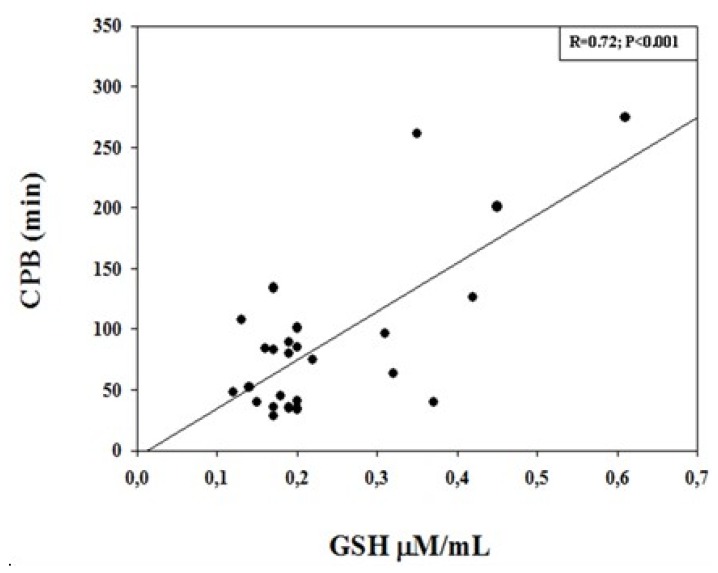
Correlation between GSH concentrations (µM/mL) measured at the end of cardiopulmonary bypass (CPB) procedure and CPB duration (min). There was a significant positive correlation (*R* = 0.72; *p* < 0.001).

**Figure 3 diagnostics-09-00118-f003:**
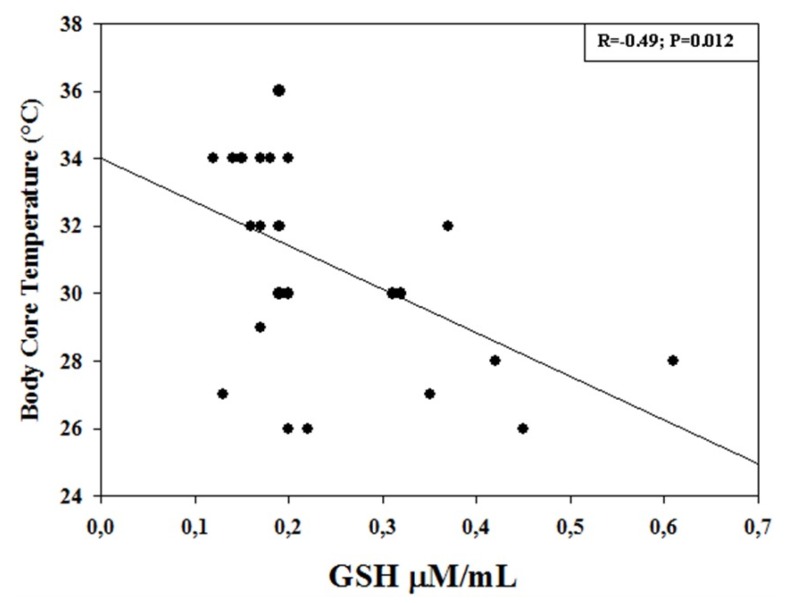
Correlation between GSH concentrations (µM/mL) measured at the end of cardiopulmonary bypass (CPB) procedure and body core temperature (°C) during CPB. There was a significant negative correlation (R = −0.49; *p* = 0.012).

**Figure 4 diagnostics-09-00118-f004:**
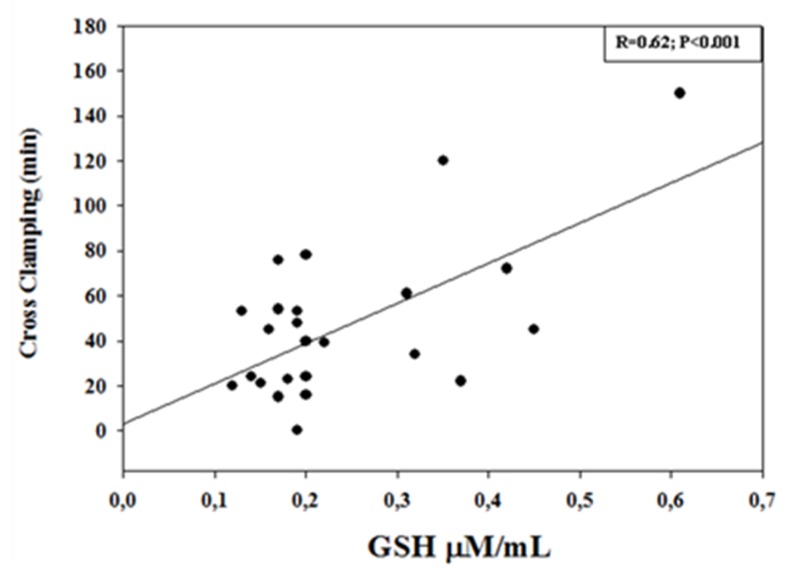
Correlation between GSH concentrations (µM/mL) measured at the end of cardiopulmonary bypass (CPB) procedure and cross-clamping duration (min). There was a significant positive correlation (R = 0.62; *p* < 0.001).

**Table 1 diagnostics-09-00118-t001:** Laboratory parameters, main interventions, and general characteristics of the children with complications of congenital heart disease (CHD) admitted into the study.

	CHD (*n* = 35)
CHD characteristics	
Tetralogy of Fallot	15
Transposition of great arteries	6
Tricuspid atresia	8
Total anomalous pulmonary venous return	6
Age (months)	30 ± 8
Weight (kg)	11 ± 2
Gender (F/M)	10/25
Laboratory parameters	
Hemoglobin (g/dL)	12.3 ± 1.2
Hematocrit (%)	35.5 ± 2.9
Platelet count (10^3^/mmc)	325 ± 102
Creatinine (mg/dL)	0.43 ± 0.25
Urea (mg/dL)	29 ± 14
LDH (UI/L)	565 ± 206
CK (UI/L)	168 ± 114
Glycaemia (mg/dL)	103 ± 12
Neurological examination	
Preoperative (normal/suspect/abnormal)	35/0/0
Postoperative (normal/suspect/abnormal)	35/0/0
Main interventions	
CPB (min)	90 ± 66
Filtration (*n*/total)	19/25
Clamping (min)	46 ± 34
Circulatory arrest (*n*/total)	2/25
Cooling (°C)	31.9 ± 3.1

Abbreviations: LDH, lactate dehydrogenase; CK, creatine kinase, CPB, cardiopulmonary bypass.

**Table 2 diagnostics-09-00118-t002:** Laboratory parameters at different monitoring time points (before the surgical procedure, T0; during the surgical procedure before CPB, T1; at the end of CPB, T2; at the end of the surgical procedure, T3; 24 h after the surgical procedure, T4) in children admitted into the study. Data are given as mean ± SD.

Parameters	T0	T1	T2	T3	T4
Hemoglobin (g/dL)	11.7 ± 2.2	11.5 ± 2.4	11 ± 2.9	11.1 ± 1.4	11.8 ± 1.5
Hematocrit rate (%)	35.9 ± 3.8	33.8 ± 5.4	32.9 ± 6.3	33.4 ± 4.2	34.5 ± 4.7
pH	7.36 ± 0.10	7.36 ± 0.10	7.38 ± 0.08	7.39 ± 0.08	7.42 ± 0.07
PaCO_2_ (mmHg)	36.9 ± 6.6	35.4 ± 3.9	34.4 ± 5.5	35.5 ± 5.1	36.2 ± 5.9
PaO_2_ (mmHg)	101 ± 37	144 ± 78 *	211 ± 101 *	163 ± 96 *	155 ± 86 *
HCO_3_ (mmol/L)	22.1 ± 3.1	22.2 ± 3.9	21.1 ± 3.2	21.2 ± 1.9	22.2 ± 1.8
BE (mmol/L)	0.2 ± 2.5	−1.8 ± 3.6	−3.0 ± 2.1	−0.5 ± 0.7	1.5 ± 1.5
SaO_2_ (mmHg)	94.9 ± 8.8	93.8 ± 7.2	97.3 ± 1.8	92.7 ± 2.6	95.3 ± 5.5
Heart rate (bpm)	104 ± 11	113 ± 14	121 ± 14	122 ± 12	125 ± 19
LA BP (mmHg)	8.0 ± 3.9	7.7 ± 4.0	9.2 ± 3.5	9.5 ± 4.2	9.3 ± 4.0
RA BP (mmHg)	9.1 ± 2.2	8.7 ± 1.6	8.8 ± 2.1	9.8 ± 2.3	10 ± 2.8
Systolic BP (mmHg)	88 ± 12	87 ± 17	86 ± 14	95 ± 15	96 ± 13
Diastolic BP (mmHg)	42 ± 11	53 ± 10	54 ± 10	56 ± 10	57 ± 11
Glycaemia (mg/dl)	103 ± 12	118 ± 11	130 ± 19	125 ± 15	119 ± 14

* *p* < 0.05 vs. T0. Abbreviations: arterial carbon dioxide partial pressure, PaCO_2_; arterial oxygen partial pressure, PaO_2_; arterial bicarbonate level, HCO_3_; base excess, BE; arterial oxygen saturation, SaO_2_; left atrium, LA; blood pressure, BP; right atrium, RA.

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
