# Peer review of "Glutathione Blood Concentrations: A Biomarker of Oxidative Damage Protection during Cardiopulmonary Bypass in Children"

_diagnostics, 2019, doi:10.3390/diagnostics9030118_

Round 1

Reviewer 1 Report

The purpose of this paper is to explore the association of glutathione levels as a marker of oxidative stress in children with critical congenital heart disease undergoing cardiopulmonary bypass for correction.  Theof Glutathione is an endogenous antioxidant that is increasingly able to be rapidly measured clinically and understanding its utility as a biomarker is compelling.  The strengths of this study were a relatively large cohort, with a consistent anesthetic technique allowing for the ability to correlate GSH levels with CPB time and other clinical characteristics

This paper has a few weaknesses:

1.  The introduction and previously known information is not presented clearly, which inhibits a reader from truly understanding its importance.  This is most striking in the second paragraph on the second page.

2.  Total glutathione was measured, while it would have been interesting (as the authors note in their introduction) to measure GSH and GSSG

3.  The paper would benefit from grammar editing as the grammar in many places makes comprehending the paper challenging

4.  The method used for glutathione quantifications more properly is termed to measure thiols in general.  One can speculate that the only nonproteic thiol that would increase substantially over a short time course, such as this, is glutathione; however, a more specific assay would be ideal.  Barring that, it would be helpful if the authors can confirm that no subjects were treated with drugs containing a thiol group.

5.  The authors at points (e.g. legend for figure 4) reference a PaO2/GSH ratio; however, all of the figures use GSH as the axis.  If PaO2 was used, thiss should be corrected in the figure axes and a reason for using this ratio should be explained -- since PaO2 increased in most patients, while GSH stayed stable I am very concerned that this ratio reflects PaO2 variation, not GSH variation.

6.  My largest concern with this study is that the authors conclude that GSH is significantly elevated postoperatively.  I don't see that demonstrated on figure 1.  It is hard to then evaluate the further claims made in this paper, because I am concerned that this assumption is not well demonstrated by this paper.  If there is a true significant elevation, the average GSH levels at each time point should be included, along with a p-value. 

7.  In the discussion on page 8, in paragraph 3 the authors enumerate a number of hypotheses as to why the GSH would be elevated.  The authors must make this clear.

Author Response

1. Please see the "cover letter" attached 

2. Please see the "cover letter" attached 

3. Please see the "cover letter" attached 

4. Please see the "cover letter" attached 

5. Please see the "cover letter" attached 

6. Please see the "cover letter" attached 

7. we enumerated the hypotheses (please find the manuscript attached). 

Reviewer 2 Report

The authors measured blood glutathione (GSH) during and post-24 h of by-pass heart surgery of children and documented the increased level of GSH which might protect tissues from oxidative damage. Definitely, this is an important piece of work which gives the glimpse of the body’s own protection mechanism kicking in to reduce the tissue damage during any invasive procedure which may change the oxygen content in the body. I recommend publishing the paper with minor modification and clearing a few ambiguities.

In the title “a Biomarker of Oxidative Damage” was actually intended to mean “a Biomarker of Oxidative Stress” or “a Biomarker of Oxidative Damage Protection”. I suggest considering the title modification accordingly. The title of the manuscript correctly described the patients as “children”. However, in the body of the manuscript, the authors refer the patients as ‘infant’ several times. In the US and some other places “infant” refers to kids less than 2-year-old. Here patients are 30 ± 8 months old. For broader readers’ convenience authors may consider changing “infant” to child(ren) throughout the MS. The time point for blood sample collection T0 is mentioned as “before the surgical procedure”. Is it before anesthesia or after? I wonder if the anesthesia process (the meds and the use of oxygen) has any effect on the level of GSH. The time point at the end of CPB (T2); at the end of the surgical procedure (T3): - During the procedure, fluid was transfused. Do the authors consider the volume increase in the circulatory system and normalized the amount of GSH/unit volume accordingly when they are comparing data with T0? Or just the directly measured amount/unit volume is reported? Authors should state that in the manuscript. Figure 1: it is very hard to tell by looking at the picture, the increment in the median GSH concentration. Please try to make the picture better to show the difference visually. Also, write the numerical values of the data used in figure 1 in the manuscript or present as a table. Fig 2 is GSH concentration versus CPB duration (T2) of all subjects. If it is true, please mention in the figure legend that CPB duration is T2. Figure 3 and 4: Figure and figure legends were interchanged. Finally, the experimental design is perfect to exclude the neurological symptom containing children. But it would be interesting to know the data of the children those showed the symptom after the procedure if it has really anything to do with their GSH level.

Author Response

We correct the term "infants" with "child/children" as requested (please find in the manuscript attached).

Fig 2: T2 is "the end of CPB" and not the "CPB duration". 

Fig 3 and 4: we corrected the figure legends (please find in the manuscript attached)

Please see the "cover letter" attached. 
